# Identity-Preserving Human Reconstruction from a Single Image via 3D Token Reasoning

## Abstract

We present the Identity-Preserving Large Human Reconstruction Model (IPRM), a feed-forward framework that reconstructs photorealistic, clothed 3D humans from a single in-the-wild image while preserving 3D identity. Recent works predominantly reason 3D structure based on 2D features, making it challenging to achieve 3D consistency while preserving the human identity in 3D space. To alleviate these challenges, IPRM anchors the monocular 3D reasoning human reconstruction by constructing a human-based 3D feature space and explicitly preserves the human identity and details by the 3D features. Specifically, we introduce an efficient and robust SMPL-based sparse voxel representation to transform 2D identity features into 3D space, categorizing them as 3D visible identity tokens and invisible tokens to be reasoned. Using these 3D tokens, an identity-aware 3D reasoning module is proposed to propagate projected 3D identity features from visible to invisible tokens, ensuring that only unobserved regions are reasoned while observed identity remains intact. Subsequently, IPRM introduces an encoder-decoder structure to decode SMPL-based 3D features into 3DGS and mesh representation, while simultaneously designing a 3D ID Adapter for identity preservation. Instead of only conditioning on 2D image tokens, this adapter utilizes 3D identity tokens extracted from a single-view branch as guidance to inject identity information at the 3D token level. Comprehensive experiments on existing benchmarks and in-the-wild data show that IPRM surpasses state-of-the-art methods in reconstruction performance, efficiency, and identity consistency.

## 1 Introduction

The pursuit of photorealistic 3D clothed human models represents a rapidly evolving field with substantial commercial impact across gaming, film production, fashion design, and AR/VR applications. Although dense-view capture systems with scene-specific reconstruction models demonstrate success in achieving this goal (Işık et al., 2023), their substantial computational and hardware requirements impede widespread deployment. Therefore, a user-friendly system is needed that can reconstruct 3D humans from any in-the-wild image in a feedforward manner

This task is challenging, where the core target lies in how to **reason the complete 3D representation from monocular input while preserving human identity** (Li et al., 2024). Recent approaches often address this issue using 2D features, either by employing multiview or video-based generation to enhance 2D view completeness (Li et al., 2024), or directly utilizing 2D image feature tokens to update 3D geometric tokens (Qiu et al., 2025a) that are sampled on parametric body models (SMPL) (Loper et al., 2015) (referred to as 2D-to-3D Token Reasoning). However, these approaches struggle to maintain multi-view consistency and fail to ensure that the reasoned 3D identity features align with the 2D image, leading to issues such as artifacts and identity drift, as shown in Fig 1.

Therefore, we propose the Identity-Preserving Large Human Reconstruction Model (IPRM), a novel framework that anchors the single-view human reconstruction process in an identity-aligned 3D SMPL feature space and, rather than only using 2D image tokens, explicitly preserves human identity through 3D visible token guidance. However, implementing this framework presents several significant challenges: 1). Recent studies (Qiu et al., 2025a; Pan et al., 2024) focusing on sampling point-features from the coarse SMPL model struggle to accurately project 2D identity features into these visible 3D points, with excessive points often compromising efficiency. 2). Existing works (Qiu et al., 2025a) directly update all 3D token features conditioned on 2D image tokens, making it difficult to preserve the projected 3D identity features. 3). The decoding process from

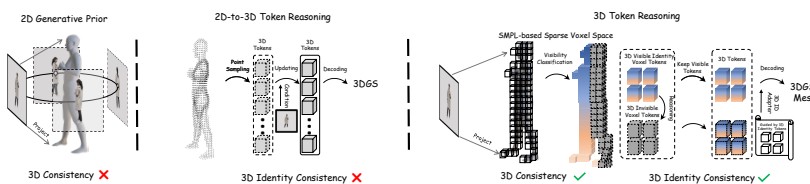

Figure 1: **Existing methods** based on 2D generative priors and 2D-to-3D Token Reasoning often over-rely on 2D feature space and fail to maintain 3D consistency and explicitly align 3D human identity with the 2D input reference. **IPRM** achieves 3D Token Reasoning from 3D visible identity tokens to 3D invisible tokens directly and preserves consistency on 3D identity-related tokens.

the SMPL-based feature space to 3D representations is often underexplored, despite its potential to introduce uncertainty in the 3D identity features.

To address these challenges, 1). We introduce a **sparse voxel representation in SMPL space** for 3D human modeling, which is more efficient compared to SMPL-point sampling and exhibits greater robustness for SMPL-induced feature projection errors. 2). We design an **identity-aware 3D reasoning module**, which includes multiple visibility mask-based self-attention blocks to achieve the transition only from 3D visible identity to invisible tokens, and an adaptive 3D Human Feature to align these 3D tokens with the human-specific domain. 3). IPRM introduces a **3D ID Adapter** based on a parallel single-view branch to maintain the consistency of identity-related tokens at the token level during the decoding process. Specifically, IPRM first projects 2D features into 3D sparse active voxels occupied by SMPL prior as well as camera Pose, and leverages SMPL's normal direction to distinguish visible and invisible voxels, thereby deriving visible identity and invisible tokens within the SMPL space. Based on these tokens, the identity-aware 3D reasoning module infers invisible tokens using visibility mask-based self-attention blocks, which can explicitly preserve 3D identity tokens unchanged. Subsequently, they are further refined by the 3D Human Feature, which includes a visible Human Feature derived from the input image and an invisible Human Feature predicted from the reasoned 3D invisible tokens to represent the entire 3D target. Finally, we train an encoder-decoder structure to decode these features from SMPL space, supporting both 3DGS and mesh representations. To further preserve identity consistency during this process, we design an additional single-view branch to predict the 3D identity condition tokens, which provides token-wise guidance to ensure identity consistency in 3D space.

In summary, IPRM is a novel 3D human reconstruction framework capable of achieving fast and high-fidelity human reconstruction from single-image features within **one second**, while utilizing minimal GPU memory. Quantitative and qualitative experiments on extensive benchmarks demonstrates the superiority of IPRM, particularly in preserving identity features. Furthermore, IPRM can be directly extended to multi-view input settings. Our main contributions are as follows:

- IPRM establishes a novel paradigm for directly reconstructing 3D humans while preserving 3D identity features via 3D token reasoning on SMPL-based 3D sparse voxel representation.

- We propose an identity-aware 3D reasoning module, which includes visibility mask-based self-attention blocks to maintain human 3D identity features consistency during the 3D reasoning process, and a 3D Human Feature for further refinement with human-specific knowledge.

- IPRM supports decoding into diverse 3D representations, including 3DGS and mesh. Additionally, we introduce a 3D ID Adapter as critical 3D guidance to mitigate identity drift at the 3D token level, significantly enhancing identity consistency throughout this process.

- IPRM achieves efficient inference of 3D human representations from image features in approximately 0.6 seconds. Comprehensive qualitative and quantitative evaluations validate the framework's superiority over existing methods and demonstrate the effectiveness of its key components in addressing identity preservation and 3D consistency.

## 2 RELATED WORKS

### 2.1 DIFFUSION-BASED SINGLE-IMAGE HUMAN RECONSTRUCTION

Reconstructing 3D humans from a single view is inherently an ill-posed problem. Therefore, a straightforward idea is to introduce 2D diffusion models for providing additional priors. Early

works (Zhang et al., 2024a; 2025b; Gao et al., 2024; Wang et al., 2024) attempt to leverage Score Distillation Sampling (SDS) (Tang et al., 2023; Huang et al., 2024), using diffusion models as auxiliary supervision to optimize 3D models, but these approaches often require time-consuming optimization. Later, leveraging advances in diffusion models, SiTH (Ho et al., 2024) address missing 3D information by dividing the task into generative hallucination and reconstruction, using diffusion models to hallucinate unseen back-view appearances from input images. Building on this idea, subsequent works adopt a similar generation-reconstruction paradigm (Chen et al., 2024; Sengupta et al., 2024; Zhang et al., 2025a). Among these works, (AlBahar et al., 2023) further focuses on preserving the shape and structural details of the underlying 3D structure, while PSHuman (Li et al., 2024) introduces an iterative Explicit Human Carving approach to achieve similar goals. In addition to multi-view generation models, HumanSplat (Pan et al., 2024) utilized pre-trained video generation models to complete continuous 3D information, while DiffHuman (Sengupta et al., 2024) extends image generation to different domains such as normal and depth maps. Although these diffusion model-based methods have achieved great success in recent years, generation in 2D space struggles to align with 3D structures, inevitably leading to 3D inconsistencies. Additionally, the multi-step nature of diffusion models inevitably sacrifices efficiency. To address these issues, IPRM introduces a novel framework that reasons invisible features directly in 3D SMPL-based feature space without diffusion processes and facilitates the transformation from SMPL space to realistic 3D representations while maintaining 3D identity consistency.

## 2.2 Large Human Reconstruction Model

Single-image 3D human reconstruction is traditionally tackled by directly regressing the parameters of a parametric body model (Kanazawa et al., 2018; Kolotouros et al., 2019) or by learning pixel-aligned implicit surfaces that capture clothing details (Saito et al., 2019; 2020; Xiu et al., 2022; 2023) (ICON, 2021; ECON, 2023). However, such methods lack explicit representations and struggle with poor geometric quality. The development of Large Reconstruction Models (LRMs) (Hong et al., 2023; Tang et al., 2024) has revitalized this field by enabling generalizable feedforward object reconstruction from a single or a small number of images. In the task of human reconstruction, several works adopt this paradigm to replace traditional reconstruction methods (Zhuang et al., 2024). Early Human-LRM (Weng et al., 2024) uses three-plane features for coarse reconstruction and improves representation with diffusion models. Human-Splat (Pan et al., 2024) utilizes generated multi-view image features to implicitly update SMPL-based Human Geometric Token features and decodes the optimized SMPL-based 3D features into 3DGS representations. Similarly, (Chen et al., 2024) adopts this paradigm and introduces additional optimization for back-view perspectives. In contrast, LHM (Qiu et al., 2025a) and its derivative work (Qiu et al., 2025b) abandon the multi-view generation process, propose a video-based training paradigm, and introduce a Multimodal Body-Head Transformer to further enhance the reconstruction quality of head tokens.

Although these methods achieve better 3D structure compared to diffusion-based approaches, they suffer from two key limitations. First, the direct update of all 3D token features based on 2D image token fails to establish explicit correspondence with the input image, resulting in the loss of identity-specific 3D features. Second, existing methods rely on collecting numerous feature points from SMPL (Qiu et al., 2025a), which not only compromises computational efficiency but also struggles to handle cases involving loose clothing that deviates from the SMPL topology. To address these challenges, IPRM adopts an SMPL-based sparse voxel representation as the foundational 3D structure. This representation can be efficiently projected from image features and subsequently decoded into 3DGS and mesh. More importantly, unlike methods that rely on 2D-3D token reasoning, we develop a novel identity-aware 3D token reasoning paradigm to directly infer invisible voxel token features from visible identity tokens while explicitly preserving 3D identity features.

## 3 Method

### 3.1 Human Parametric Model

SMPL (Loper et al., 2015) and its variants, SMPL-X (Pavlakos et al., 2019), serve as essential parametric body model priors in human reconstruction tasks, establishing a coarse 3D geometr that facilitates subsequent reconstruction processes. In this work, we utilize SMPL-X to establish SMPL-based sparse voxel representations. The SMPL-X model employs shape parameters and

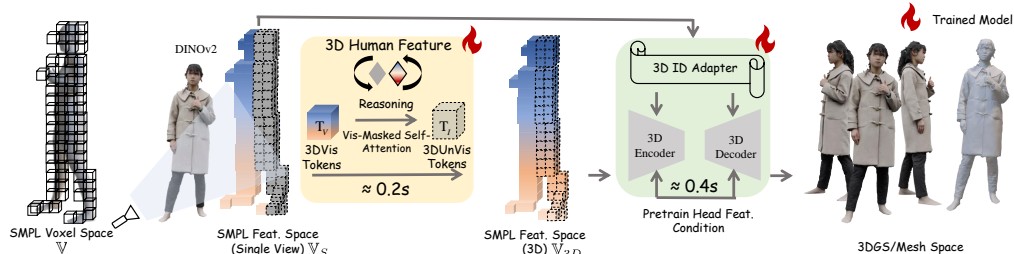

Figure 2: **Overview of IPRM's Inference Process**: After constructing a voxel space based on SMPL and projecting image features into it, IPRM classifies all voxels into visible $\mathbf{T}_V$ and invisible $\mathbf{T}_I$ categories, leveraging visible voxels to infer invisible features and obtain 3D features. Subsequently, these SMPL-based 3D features are processed through a 3D ID Adapter and encoder-decoder architecture to achieve identity-preserving 3DGS/mesh representations.

pose parameters to represent human body configurations, which can be directly converted into mesh representations in 3D space to estimate occupied voxels and the visibility mask in IPRM.

## 3.2 IPRM FRAMEWORK

Given an in-the-wild human image $\mathbf{I}$, our goal is to reconstruct a realistic 3D human representation in one second while explicitly preserving the human identity of the input. To achieve this, we propose IPRM, a feed-forward framework that projects the input image into visible identity 3D token features and leverages them to explicitly reason about invisible 3D token features, thereby enabling 3D human reconstruction from a single view. The entire pipeline is illustrated in Fig. 2.

Previous 2D-3D Token Reasoning works (Qiu et al., 2025a; Pan et al., 2024) primarily rely on sampling points directly on the SMPL surface. This approach involves extensive sampled points and depends heavily on the coarse SMPL prior, leading to inefficiencies and challenges in achieving accurate pixel-aligned feature projection due to inaccuracies in SMPL geometry. Inspired by Trellis (Xiang et al., 2025), IPRM adopts an SMPL-based sparse voxel as the foundational representation. By operating on active voxels rather than individual surface points, this method significantly improves efficiency and mitigates the impact of 2D-3D projection errors. Combined with the subsequent encoder and decoder, it enables conversion from the SMPL feature space to 3DGS and mesh.

After projecting image features into the SMPL-based voxel space, referred to as the SMPL Feature Space (Single View), we propose an identity-aware 3D reasoning module. This module leverages projected visible voxel tokens (also referred to as identity tokens) to infer invisible 3D tokens while preserving identity consistency in the 3D space, resulting in the SMPL Feature Space (3D). During the encoding-decoding process from SMPL-based feature space to 3DGS or mesh, IPRM introduces a 3D ID Adapter, which utilizes single-view 3D identity features from the condition branch as guidance to mitigate identity drift in a token-by-token manner.

## 3.3 SMPL-BASED SPARSE VOXEL TOKENS

Sparse voxel is an efficient 3D representation designed to coarsely capture the occupied regions of a 3D space surrounding an object (Gao et al., 2023; Lu et al., 2024). Unlike Trellis (Xiang et al., 2025), which directly operates in real geometric space, our sparse voxel is adapted to the SMPL space by identifying active voxels based on whether they are occupied by the SMPL geometry. Instead of sampling exact points on the SMPL surface, this voxel-based representation captures the 3D space surrounding SMPL and outlines the human's coarse geometry on the 3D grid (Xiang et al., 2025), while being less sensitive to feature projection inaccuracies on the SMPL surface. Correspondingly, to enable high-quality conversion from the SMPL space to 3DGS and mesh, the specific encoder-decoder structure is trained to further enhance robustness to approximate projected voxel features. The effectiveness of this representation is discussed in Sec. 4.3 and Fig. 7.

Based on the SMPL-based sparse voxel $\mathbb{V}$ and input features $\mathbf{F} = \mathrm{DINOv2}(\mathbf{I})$, we project $\mathbf{F}$ into the 3D voxel space and serialize them as tokens $\mathbf{T} = \{(z_i, f_i)\}_{i=0}^{V}$, where $z_i \in \{0, 1, \ldots, N-1\}^3$ is the positional index of the active voxels and $f_i \in \mathbb{R}^C$ represents the projected feature from $\mathbf{F}$. We refer to it as the SMPL feature space (Single view), denoted as $\mathbb{V}_S$. Subsequently, IPRM

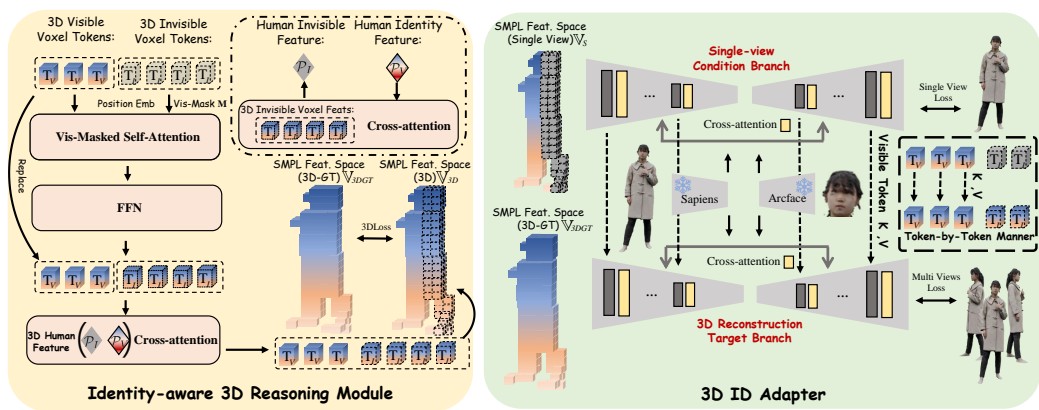

Figure 3: Given voxel tokens, the **identity-aware 3D reasoning module** uses multiple visibility mask-based self-Attention blocks and a 3D Human Feature to update invisible voxel token features and refine 3D Tokens for human-specific knowledge alignment, respectively. During the decoding of these SMPL-based 3D features to 3DGS/mesh, the **3D ID Adapter** designs a single-view branch to predict single-view 3D identity features, which serve as guidance to preserve the identity of 3D representations in a token-by-token manner.

leverages the relationship between the SMPL-based voxel normals (*Norm*) and viewpoints $\mathbf{P}$ to classify these tokens into visible tokens $\mathbf{T}_V$, i.e., identity tokens, and invisible tokens $\mathbf{T}_I$, i.e., tokens to be reasoned and obtain a visibility mask $\mathbf{M}$ based on the token index $z$.

### 3.4 IDENTITY-AWARE 3D REASONING MODULE

Given $\mathbf{T}_V$, the IPRM is tasked with inferring $\mathbf{T}_I$ while preserving the integrity and consistency of the known identity features $\mathbf{T}_V$. To address this objective, we introduce the identity-aware 3D reasoning module, which incorporates a visibility-mask-guided self-attention mechanism alongside a 3D Human Feature. The integration of these components enables effective reasoning and facilitates the transformation from $\mathbb{V}_S$ to SMPL feature space (3D), referred to as $\mathbb{V}_{3D}$.

To handle $\mathbf{T}_I$ and $\mathbf{T}_V$, IPRM first incorporates sinusoidal positional encodings by voxel positions and concatenates them token-wise, which are subsequently processed through self-transformer blocks. Given the visibility mask $\mathbf{M}$, the visibility mask-based self-attention blocks are then employed to reason over $\mathbf{T}_V$ while explicitly preserving $\mathbf{T}_I$ unchanged. This process enables reasoning from known regions to unknown regions, but it struggles to align with human-specific knowledge. Therefore, we introduce an additional human-based condition to further refine these 3D tokens.

Prior approaches (Qiu et al., 2025a) typically depend on human-specific encoders (Khirodkar et al., 2024), leveraging a single-view human feature as conditioning, referred to as $\mathcal{P}_V$, to accomplish this goal. However, relying exclusively on this 2D Feature to guide the update of all tokens $\mathbf{T}$ proves insufficient, as 2D features extracted from limited viewpoints inherently struggle to align with the comprehensive 3D spatial representation. Therefore, we propose a 3D Human Feature, which introduces an additional invisible Human Feature $\mathcal{P}_I$ representing the invisible 3D information that is not captured from the given viewpoint. Subsequently, the IPRM leverages the concatenation ($||$) of both components as a condition to collectively guide the updating of 3D tokens. To obtain $\mathcal{P}_I$, we utilize the reasoned $\mathbf{T}_I$ as a condition to update $\mathcal{P}_V$, which is achieved through the cross-attention blocks, enabling the 3D Human Feature to transition from visible to invisible regions, as in Fig. 3.

$$\mathcal{P}_I = \text{Cross-Attention}(\mathcal{P}_V; \mathbf{T}_V), \quad \mathbf{T} = \text{Cross-Attention}(\mathbf{T}_V || \mathbf{T}_I; \mathcal{P}_V || \mathcal{P}_I), \quad (1)$$

### 3.5 3D ID ADAPTER

After achieving the transformation from SMPL-based single-view features to 3D features, IPRM aims to decode these 3D voxel features $\mathbb{V}_{3D}$ into 3DGS or mesh representations. This process has been discussed in existing works (Qiu et al., 2025a; Xiang et al., 2025), but they often overlook the potential identity drift introduced during this step. Such a drift may arise due to the uncertainty

Table 1: Comparisons of methods on two benchmark (Qiu et al., 2025a; Li et al., 2024) on Synthetic Data and THuman2.1 for 3DGS reconstruction, with inference time and memory usage.

| Methods | Synthetic Data | | | | | THuman2.1 | | | Inference | |
|---|---|---|---|---|---|---|---|---|---|---|
| | PSNR↑ | SSIM↑ | LPIPS↓ | PSNR(I)↑ | FC↓ | PSNR↑ | SSIM↑ | LPIPS↓ | Time | Memory |
| GTA | 17.03 | 0.919 | 0.09 | 17.66 | 0.051 | 19.61 | 0.834 | 0.10 | 0.68s | ≈8GB |
| SIFu | 16.68 | 0.917 | 0.09 | 19.22 | 0.060 | 19.44 | 0.831 | 0.10 | 0.65s | ≈9GB |
| DreamGaussian | 18.54 | 0.917 | 0.08 | 19.47 | 0.056 | - | - | - | 2 min | ≈8GB |
| SITH | - | - | - | - | - | 18.46 | 0.820 | 0.10 | 45.12s | ≈21GB |
| PSHuman | 17.56 | 0.921 | 0.08 | 21.44 | 0.037 | 20.85 | 0.864 | 0.08 | 1 min | ≈40GB |
| Trellis* | 21.67 | 0.921 | 0.05 | 21.99 | 0.058 | 21.33 | 0.886 | 0.06 | 4.20s | ≈11GB |
| LHM-0.5B | 25.18 | 0.951 | **0.03** | 26.64 | 0.035 | - | - | - | 2.01s | ≈18GB |
| Ours | **27.11** | **0.954** | **0.03** | **28.86** | **0.031** | **26.87** | **0.949** | **0.05** | **0.61s** | ≈9GB |

challenge in directly decoding 3D tokens, defined within the SMPL space, into 3D representations. A popular approach is to incorporate a 2D image condition to guide the 3D token decoding.

We explore the integration of human features (Khirodkar et al., 2024) and face features (Deng et al., 2019) as conditioning inputs for the cross-attention layers to guide the encoder-decoder. Nevertheless, experiments demonstrate that this approach yields only modest improvements, potentially due to the challenge of establishing explicit correspondence between the 2D tokens and the 3D tokens, as aforementioned. To address this issue, IPRM further introduces 3D identity condition tokens, which not only directly guide token decoding in a token-by-token manner but also achieve better alignment within the same feature domain. To obtain the 3D identity condition tokens, we design an additional single-view condition branch that operates in parallel with the target branch. Distinct from the target branch (reconstruction branch), which takes $\mathbb{V}_{3DGT}$ as input and focuses on reconstructing 3D representations from SMPL-based features, the condition branch processes $\mathbb{V}_S$ as input with the objective of reconstructing the input or identity image only.

Through optimizing condition branch by the input image, it acquires the capability to infer single-view 3D identity features from a 2D input image, serving as critical guidance for the target branch. Motivated by the design of ReferenceNet (Hu, 2024), we replace the key and value in each self-attention layer of the target branch with the corresponding key and value from the condition branch, while preserving the query from the target branch. In addition, the objective of the 3D ID Adapter is to leverage 3D identity features to ensure the identity consistency. Therefore, this 3D token-wise guidance is applied exclusively to the visible identity tokens, while the reasoned invisible tokens retain the original key and value from the target branch.

## 3.6 Training Strategy

IPRM adopts a two-stage training paradigm to separately train the identity-aware 3D reasoning module and the encoder-decoder with the 3D ID Adapter. In the first training stage, we take images and SMPL as inputs to predict SMPL-based 3D voxel features $\mathbb{V}_{3D}$ from the SMPL-based single-view feature space $\mathbb{V}_S$. This process is supervised in the GT-3D feature using MSE loss. To obtain the SMPL-based GT 3D voxel features $\mathbb{V}_{3DGT}$, we project and aggregate the features from all viewpoints into the 3D SMPL voxel space, as in appendix A. In the second stage, we fine-tune the pre-trained encoder-decoder (Xiang et al., 2025) without sampling, which is designed to directly decode into 3DGS or mesh representations from SMPL-based $\mathbb{V}_{3DGT}$. Additionally, we jointly train the model with image-based cross-attentions and a 3D ID Adapter to preserve identity features. Specifically, the 3DGS-decoder maps 3D voxel feature into Gaussian components with position offsets, colors, scales, opacities, and rotations, optimized using $L_1$, D-SSIM, and LPIPS losses, while for meshes, 3D features are transformed into signed distance values for voxel vertices, with outputs extracted from isosurfaces and optimized using $L_1$ loss on depth and normal maps. For the condition branch, we supervise it only through a single view under the input view.

## 4 Experiments

### 4.1 Experiments Detail

**Dataset and Evaluation Metrics**: We introduce Human4Dit (Shao et al., 2024), Thuman2.1 (Yu et al., 2021), 2K2K (Han et al., 2023), and CustomHumans (Ho et al., 2023) for training and evaluat-

ing the IPRM. To ensure fairness, we follow the evaluation benchmarks of LHM (Qiu et al., 2025a) and PSHuman (Li et al., 2024) by using the same test cases and viewpoints for assessment while ensuring that these evaluation cases are excluded from the training set. To quantitatively evaluate the performance of 3DGS rendering, we introduce traditional metrics such as PSNR, SSIM, and LPIPS to assess the performance of 3D human reconstruction. Additionally, we introduce Input-view PSNR, referred to as PSNR (I), and Face Consistency (Qiu et al., 2025a) to further evaluate the ability of IPRM to preserve identity. To assess the quality of mesh reconstruction, we employ three primary metrics: one-directional point-to-surface (P2S), L1 Chamfer Distance (CD), and Normal Consistency (NC) (Li et al., 2024). More details are discussed in appendixA.

**Preprocessing Pipeline**: Given an in-the-wild image, we first utilize Segment Anything (Kirillov et al., 2023) to segment the foreground human. Subsequently, we generate square and centralized input images and estimate their SMPL-X following SiTH (Ho et al., 2024) for reconstruction.

**Training Configuration**: IPRM adopts a two-stage training paradigm. Specifically, the identity-aware 3D reasoning module is trained from scratch with an initial learning rate of $2 \times 10^{-3}$ over 200,000 iterations. In contrast, the encoder-decoder is fine-tuned from a pre-trained model (Xiang et al., 2025), where cross-attention layers are added and initialized. This stage is optimized with an initial learning rate of $5 \times 10^{-4}$ over 100,000 iterations. More details are discussed in appendixA.

## 4.2 COMPARATIVE EXPERIMENTS

Here, we primarily follow three recent and widely adopted benchmarks for conducting comparative experiments: 1) approaches leveraging 2D generative priors (Li et al., 2024), 2) 2D-to-3D Token Reasoning methods (Qiu et al., 2025a), as Fig. 1, and 3) general 3D generation models (Xiang et al., 2025) fine-tuned on human data. To ensure a rigorous and fair comparison, all experiments are conducted on identical test cases (excluded from the training data) and under consistent viewpoints.

### 4.2.1 QUANTITATIVE EVALUATION

**Performance Comparison**: As evidenced by the experimental results in Tab 1 and Tab 4, our proposed IPRM achieves substantial improvements across all evaluation metrics compared to existing state-of-the-art methods. These performance gains can be primarily attributed to the framework's ability to perform 3D token reasoning directly in 3D space, enabling superior 3D consistency, particularly when compared to approaches based on 2D generative models. As reflected in the PSNR(I) and FC metrics, IPRM demonstrates improved preservation of human identity compared to LHM (Qiu et al., 2025a). This improvement stems from our SMPL-based sparse voxel and identity-aware 3D reasoning module, which achieves 3D tokens with 2D feature alignment and facilitates the transition from identity 3D tokens to invisible tokens. Compared to Trellis fine-tuned on human datasets, IPRM achieves greater consistency with the original images and excels in capturing fine-grained details, such as facial features, owing to the integration of SMPL priors and the 3D ID adapter. Furthermore, as shown in Tab. 2, IPRM significantly outperforms most existing methods in mesh reconstruction, delivering results comparable to PSHuman (Li et al., 2024). This achievement is particularly notable, as PSHuman relies on iterative optimization for each individual case, whereas IPRM operates within a feed-forward paradigm, offering both efficiency and effectiveness.

**Efficiency and Memory Usage**: Compared to existing algorithms, IPRM achieves superior computational efficiency, characterized by faster inference speed and reduced memory consumption, as in Tab. 1. This efficiency is primarily attributed to the framework's avoidance of the iterative denoising process of diffusion models and its adoption of an SMPL-based sparse voxel representation.

### 4.2.2 QUALITATIVE EVALUATION

As corroborated by the quantitative results, IPRM also demonstrates remarkable performance in the qualitative evaluations presented in Fig. 4. Methods based on 2D generative priors, such as PSHuman (Li et al., 2024), suffer from significant 3D inconsistency issues, leading to severe artifacts like broken fingers and distorted geometry, as shown in the facial regions in the first row and the hand regions in rows 2-3. Similarly, 2D-to-3D Token Reasoning approaches like LHM (Qiu et al., 2025a), exhibit noticeable inconsistencies in preserving input pose and identity, as shown in the facial regions in rows 1-2 and the hand regions in the third row. In contrast, IPRM effectively addresses

Table 2: Quantitative comparison for mesh.

| Method | Opt | THuman2.1 | | |
| | | Cham. Dist ↓ | P2S ↓ | NC ↑ |
|---|---|---|---|---|
| ICON | × | 0.6146 | 0.5934 | 0.8493 |
| ECON | × | 0.6725 | 0.6331 | 0.8362 |
| GTA | × | 0.5791 | 0.5587 | 0.8491 |
| SIFU | × | 0.5754 | 0.5576 | 0.8500 |
| HiLo | × | 0.5977 | 0.5892 | 0.8405 |
| SITH | × | 0.6474 | 0.5810 | 0.8264 |
| PSHuman | ✓ | **0.4399** | **0.4077** | 0.8504 |
| **Ours** | × | 0.4451 | 0.4332 | **0.8507** |

Table 3: Application on more inputs.

| Method | THuman2.1 | | |
| | PSNR ↑ | SSIM ↑ | LPIPS ↓ |
|---|---|---|---|
| Single View (Front image) | 26.87 | 0.949 | 0.05 |
| Two Views (+Back Image) | 27.31 | 0.953 | 0.04 |
| All Views | 27.77 | 0.961 | 0.03 |

Table 4: Comparison on CustomHuman.

| Method | CustomHuman | | |
| | PSNR ↑ | SSIM ↑ | LPIPS ↓ |
|---|---|---|---|
| SiTH | 27.07 | 0.938 | 0.07 |
| LHM-0.5B | 28.31 | 0.953 | 0.05 |
| Ours | 29.41 | 0.971 | 0.04 |

both challenges within a unified framework, delivering superior 3D consistency and maintaining identity fidelity through the proposed identity-aware 3D reasoning module and 3D ID Adapter. Notably, IPRM excels in preserving fine-grained details in complex regions such as the hands and face, where other methods often introduce distortions or compromise structural integrity. As illustrated in Fig. 5, we provide a qualitative comparison of IPRM for mesh reconstruction. While PSHuman (Li et al., 2024) relies on iterative optimization for each individual case, IPRM achieves more complete reconstruction results (e.g., facial regions) through a more efficient feed-forward paradigm, which benefits from the 3D consistency of IPRM. Furthermore, in comparison to GTA (Zhang et al., 2023) and SIFU (Zhang et al., 2024b), IPRM produces smoother surfaces and more precise geometry, demonstrating its robustness in capturing intricate 3D human identity and ensuring 3D consistency. More cases are discussed in appendix B.

## 4.3 ABLATION STUDY

To analyze the functional mechanisms of the SMPL-based sparse voxel, identity-aware 3D reasoning module, and 3D ID Adapter in IPRM, the ablation study is shown in Tab. 5.

Table 5: Componential ablation study.

| Method | Synthetic Data | | |
| | PSNR(I) | SSIM | LPIPS |
|---|---|---|---|
| SMPL-based Sparse Voxel | | | |
| w/o Feature Project | 17.01 | 0.911 | 0.10 |
| Identity-aware 3D Reasoning Module | | | |
| w/o Visibility Mask | 27.92 | 0.951 | 0.04 |
| w/o 3D Human Feature | 28.66 | 0.953 | 0.03 |
| 3D ID Adapter | | | |
| w/o 2D Attention | 28.72 | 0.954 | 0.03 |
| **Ours** | **28.86** | **0.954** | **0.03** |

**SMPL-based Sparse Voxel:** Beyond its contribution to higher efficiency, we further validate the impact of the SMPL-based sparse voxel representation on overall performance enhancement. Unlike existing approaches that directly sample point features on SMPL surfaces (Qiu et al., 2025a), the sparse voxel defines features in the 3D space surrounding the SMPL geometry. This design enhances robustness against the coarse geometric priors of SMPL, leading to more reliable reconstruction for loose clothing, as Fig. 6. Moreover, projecting features into 3D space introduces significant improvements to IPRM, enabling more accurate and identity-aligned reconstructions.

**Identity-aware 3D Reasoning Module:** The visibility mask proves to be critical for identity preservation, as it explicitly retains identity-relevant tokens when reasoning about invisible tokens. Furthermore, the 3D Human Feature significantly enhance the results by leveraging human-specific features to facilitate global optimization of 3D tokens, leading to improved reconstruction quality.

**3D ID Adapter:** Experimental results demonstrate that utilizing 2D images as conditioning inputs provides some benefit in maintaining identity information during the encoder-decoder process; however, the impact remains limited. This highlights the challenge of establishing explicit correspondences between 2D image tokens and 3D tokens. In contrast, our proposed 3D ID Adapter exhibits a significantly greater impact on performance, which is also demonstrated in Fig. 7. This improvement can be attributed to the single-view condition branch in effectively preserving identity features, as well as the enhanced token-to-token correspondence achieved through these 3D identity features, which better align with the 3D reasoning process.

**More Inputs:** IPRM is inherently capable of adapting to any number of input images. To further explore its potential, we extended IPRM to multi-input scenarios in Tab. 3. Experimental results

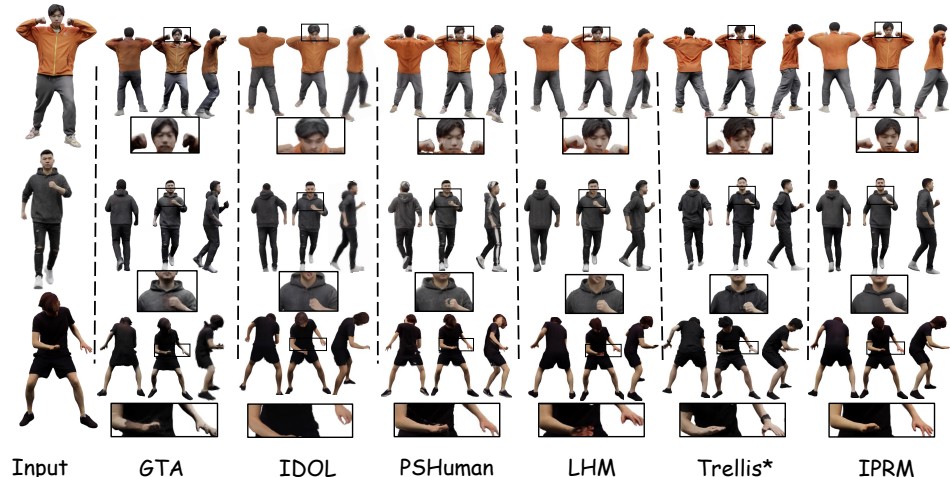

Figure 4: Qualitative comparison with recent SOTA works on test sets and in-the-wild data.

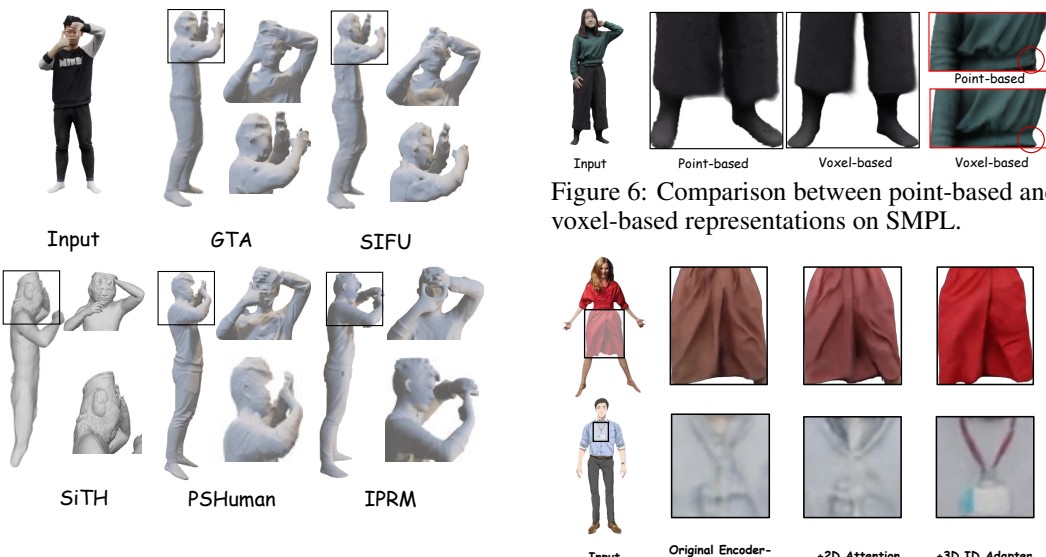

Figure 5: Qualitative comparison of IPRM with recent SOTA works on mesh.

Figure 6: Comparison between point-based and voxel-based representations on SMPL.

Figure 7: Ablation study for 3D ID Adapter.

clearly demonstrate that as the number of visible tokens increases, the prediction performance improves significantly, highlighting the model's ability to leverage additional view information for enhanced reconstruction accuracy.

## 5 CONCLUSION

This paper presented IPRM, a feed-forward framework 3D human reconstruction from single images. Through explicit 3D tokens reasoning about invisible 3D regions from visible identity features in SMPL-based sparse voxel space, IPRM achieves 3D-consistent and identity-preserving reconstruction. Our key technical contributions–the SMPL-based sparse voxel representation, identity-aware 3D reasoning module, and the 3D ID Adapter–work synergistically to preserve 3D visible identity cues while reasoning about only unobserved regions. Extensive experiments demonstrate that IPRM achieves state-of-the-art performance across multiple benchmarks, substantially surpassing existing methods in geometry accuracy, texture fidelity, and identity similarity, while operating significantly faster than prior work with minimal memory requirements.

**Limitation:** Although IPRM preserves identity, the sparse voxel representation limits fine detail reconstruction, which could be improved through future post-processing optimization.

## REPRODICIBILITY STATEMENT

We have made every effort to ensure our work is reproducible. Our experiments are conducted on the public Trellis benchmarks. The methodology for constructing the IPRM is detailed in Sec.3, and the implementation details are provided in Sec.4. To further facilitate replication, we provide comprehensive training configurations in appendix A. The source code for our framework and experiments will be made publicly available upon publication.

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

## A    EXPERIMENTAL DETAILS

### A.1    TRAINING AND EVALUATION DATASET

During training, we use partial rendered data from Human4Dit (Shao et al., 2024), Thuman2.1 (Yu et al., 2021), and CustomHumans (Ho et al., 2023), with the remaining data designated as evaluation datasets, following PSHuman (Li et al., 2024) and LHM (Qiu et al., 2025a), where Synthetic Data includes 2K2K (Han et al., 2023) and Human4Dit (Shao et al., 2024). Each case is uniformly rendered from 30 viewpoints in Blender following Trellis (Xiang et al., 2025) for training, while during evaluation, the LHM benchmark uses the same rendering viewpoints for assessment, and the PSHuman benchmark samples only four viewpoints $\{0°, 90°, 180°, 270°\}$.

## A.2 TRAINING AND EVALUATION DETAILS

After obtaining the rendered data, we can leverage them to train the 3DGS-based encoder-decoder structure and 3D ID Adapter, and utilize the GT Human Mesh along with its corresponding normals to train the mesh-based structure. To train IPRM, we reproject the multi-view rendered features back to the 3D SMPL voxel space and average them within each smpl-based voxel (Xiang et al., 2025) to obtain the GT multi-view SMPL feature space $\mathbb{V}_{3DGT}$. During inference, **we do not use** $\mathbb{V}_{3DGT}$. The IPRM can directly construct $\mathbb{V}_S$ from a single in-the-wild image and its corresponding SMPL, which is used to infer invisible tokens by the identity-aware 3D reasoning module, obtaining $\mathbb{V}_{3D}$. Finally, $\mathbb{V}_{3D}$ can be decoded into 3DGS or mesh. Additionally, in comparative experiments, we use the same method to process human data for fine-tuning Trellis (Xiang et al., 2025), including its diffusion model structure and encoder-decoder structure.

## A.3 TRAINING CONFIGURATION

All models are trained using the AdamW optimizer and conducted on 4 A6000 GPUs with a batch size of 8, requiring approximately two days for completion. The entire training framework is modified based on the open-source TRELLIS project (Xiang et al., 2025). The key difference is that we do not employ any diffusion model processes, but rather achieve single-view to 3D feature conversion solely through 3D token reasoning. In the identity-aware 3D reasoning module, we employ 4 visibility mask-based self-attention blocks in conjunction with 4 cross-attention blocks to facilitate 3D token reasoning. This configuration is designed to strike an optimal trade-off between computational efficiency and model performance. For the encoder-decoder architecture, we adopt the original Trellis (Xiang et al., 2025) structure, with the key difference being the introduction of cross-attention layers based on 2D image tokens in each self-attention block, guided by 3D identity features from the single-view condition branch. For 3DGS, we fine-tune both the encoder and decoder, while for mesh reconstruction, we only fine-tune the decoder based on the 3DGS-based encoder.

## B MORE QUALITATIVE COMPARISONS

Here we present more cases comparing IPRM with recent work, as shown in Fig. 8-10. Consistent with previous findings, IPRM demonstrates superior rendering results for the decoded 3DGS, particularly in identity features including pose, geometry, and appearance. This is attributed to the advantage of 3D Token Reasoning in preserving visible identity features. The same conclusion can be drawn for mesh reconstruction, where our method achieves more complete geometry compared to PSHuman (Li et al., 2024), which often suffers from missing parts such as hands and feet due to difficulties in maintaining 3D consistency.

## C THE USE OF LARGE LANGUAGE MODELS

During the writing process of this paper, we utilized large language models to enhance the manuscript quality, including employing large language models to correct grammatical errors and modify wording and expressions to achieve a more formal and academic style.

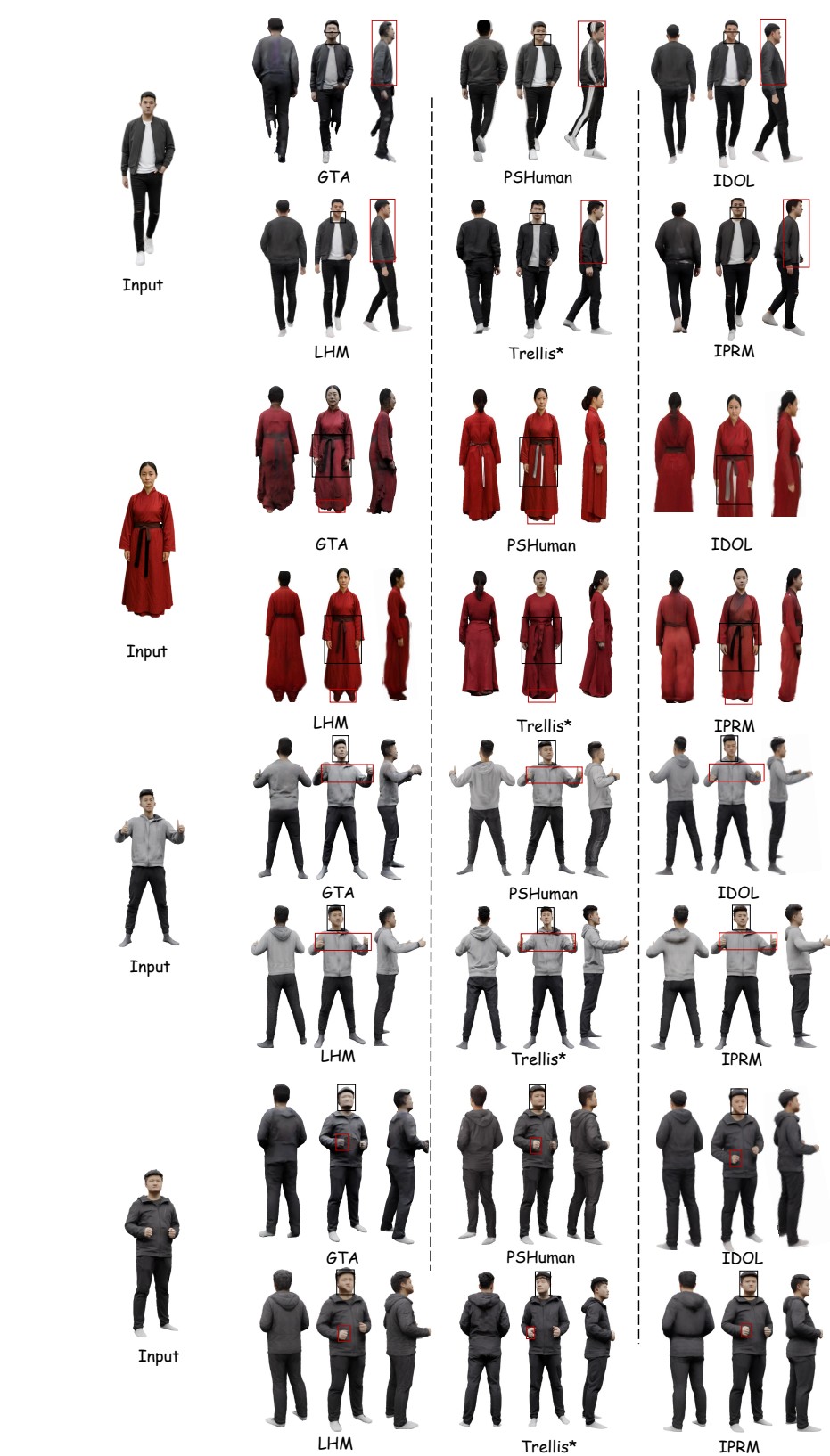

Figure 8: Qualitative comparison of IPRM with recent SOTA works.

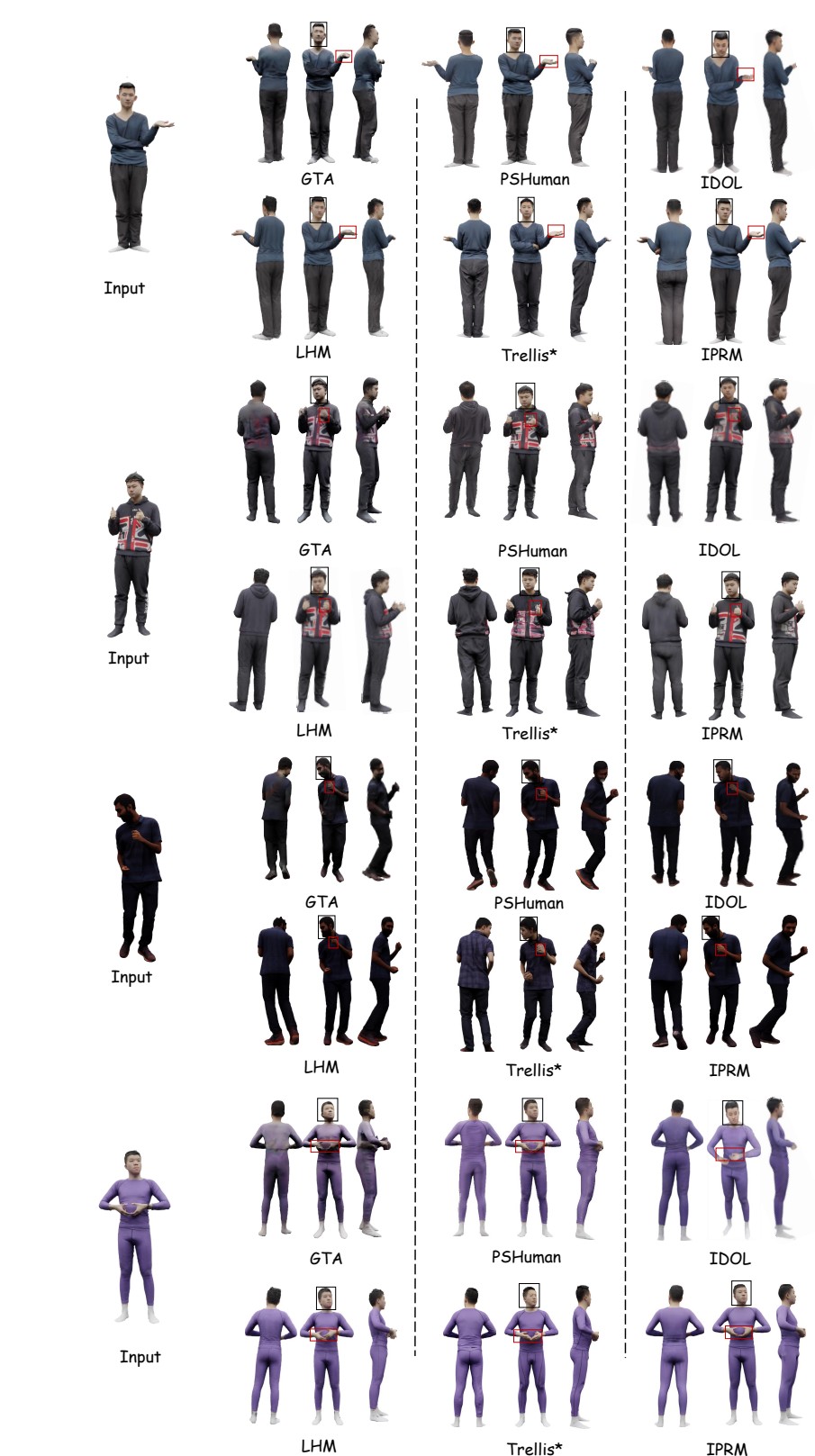

Figure 9: Qualitative comparison of IPRM with recent SOTA works.

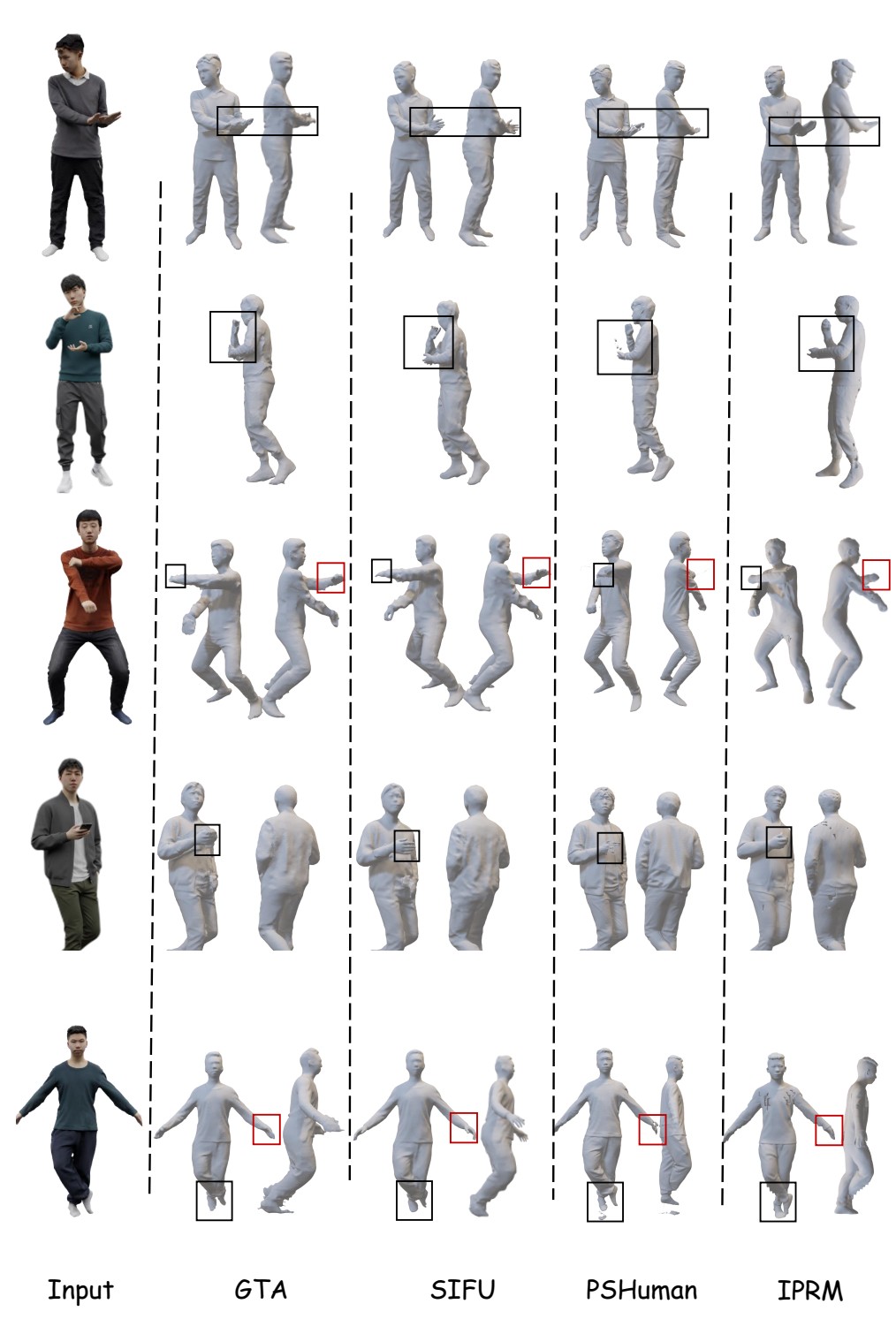

Figure 10: Qualitative comparison of IPRM with recent SOTA works.

