# OpenReview forum: "Identity-Preserving Human Reconstruction from a Single Image via Explicit 3D Reasoning"
_ICLR.cc/2026/Conference — ICLR 2026 Conference Withdrawn Submission_

### Official Review · Reviewer_bLuU · 2025-10-25

**Soundness:** 3
**Presentation:** 3
**Contribution:** 3
**Rating:** 4
**Confidence:** 4

**Summary:**

This paper presents the Identity-Preserving Large Human Reconstruction Model (IPRM), a feed-forward framework that reconstructs clothed 3D humans from a single in-the-wild image. IPRM anchors the monocular 3D reasoning human reconstruction by constructing a human-based 3D feature space and explicitly preserves the human identity and details by the 3D features. Specifically, it introduce a SMPL-based sparse voxel representation to transform 2D identity features into 3D space, categorizing them as 3D visible identity tokens and invisible tokens to be reasoned. Using these 3D tokens, an identity-aware 3D reasoning module is proposed to propagate projected 3D identity features from visible to invisible tokens. Then, IPRM introduces an encoder-decoder structure to decode SMPL-based 3D features into 3DGS and mesh representation, and designs a 3D ID Adapter for identity preservation. Experiments on existing benchmarks and in-the-wild data show that IPRM outperforms state-of-the-art methods.

**Strengths:**

- This paper introduces a method for directly reconstructing 3D humans while preserving 3D identity features via 3D token reasoning on SMPL-based 3D sparse voxel representation.
- It proposes an identity-aware 3D reasoning module, which includes visibility mask-based self-attention blocks to maintain human 3D identity features consistency during the 3D reasoning process, and a 3D Human Feature for further refinement with human-specific knowledge.
- IPRM supports decoding into diverse 3D representations, including 3DGS and mesh. Additionally, it introduce a 3D ID Adapter as critical 3D guidance to mitigate identity drift at the 3D token level, enhancing identity consistency throughout this process.
- IPRM achieves efficient inference of 3D human representations from image features in approximately 0.6 seconds. Qualitative and quantitative evaluations validate the framework’s effectiveness over existing methods.

**Weaknesses:**

- This method relies on the sparse voxel representation for feature projection and 3D reasoning. However, the paper does not specify the chosen voxel grid resolution nor provide a comprehensive ablation study on how this critical hyper-parameter affects reconstruction quality, memory consumption, and inference speed.
- The submission lacks essential validation in the form of multi-view rendering videos (e.g., 360-degree rotations). While static novel view images are provided, they are insufficient to conclusively demonstrate the robustness. This makes me feel less confident about the effectiveness of the method.
- The primary contribution of this work is stated as improving identity preservation. However, the qualitative comparisons presented in the supplementary material (e.g., Figure 8, 2nd row) suggest that existing methods like PSHuman and LHM appear visually superior or more accurate in preserving facial identity than the proposed IPRM.
- In Identity-aware 3D Reasoning Module, instead of using self-attention with mask, how about using cross-attention to query features from visible tokens to invisible tokens.
- The ablation study in Table 5 indicates that the inclusion of the dedicated 3D ID Adapter provides only marginal improvements in standard reconstruction metrics (PSNR: 28.66 vs. 28.96; SSIM: 0.953 vs. 0.954) over the baseline. Please clarify.

**Questions:**

1. Clarity on Voxel Representation and Efficiency
2. Validation of 3D Plausibility
3. Justification of Visibility Mask-based Self-Attention
4. Addressing Identity Preservation Discrepancy
5. Alternative 3D Reasoning Architectures
6. Justifying the 3D ID Adapter
See weaknesses for details.

---

### Official Review · Reviewer_apBk · 2025-10-30

**Soundness:** 2
**Presentation:** 2
**Contribution:** 2
**Rating:** 4
**Confidence:** 5

**Summary:**

This paper introduces IPRM (Identity-Preserving Human Reconstruction Model), a feed-forward framework that reconstructs clothed 3D humans from a single in-the-wild image while aiming to preserve identity. Unlike prior approaches that mainly rely on 2D features, IPRM uses a SMPL-based sparse voxel representation to project 2D identity cues into 3D space. It distinguishes between visible tokens (identity-preserving) and invisible tokens (to be reasoned), and applies an identity-aware reasoning module together with a 3D ID Adapter to prevent identity drift during decoding. Experiments on benchmarks such as THuman2.1 and CustomHuman demonstrate improvements over baselines like PSHuman, LHM, and Trellis, reporting stronger identity preservation and higher efficiency

**Strengths:**

The design of visible/invisible token separation and the 3D ID Adapter provides a clear mechanism to address identity drift, which is a common problem in this area.

**Weaknesses:**

1. Unclear Robustness to SMPL Errors

The method heavily depends on SMPL estimation, but the robustness to inaccurate SMPL poses is not systematically studied. It is also unclear in the experiments whether SMPL ground truth or estimated poses were used at test time.

2. Invisible Token Dependency on SMPL Geometry

The visible/invisible token split is derived from SMPL geometry. This could fail for subjects with loose or complex clothing that deviates substantially from SMPL, raising doubts about generalization. It is also unclear whether the proposed system would work if the input image is truncated or occluded by an object or other humans.

3. Limited Qualitative Evidence

Qualitative comparisons are shown with very small image sizes, without zoom-ins on faces. This makes it hard to judge whether identity is truly preserved or whether artifacts remain. No video results are provided, so multi-view or 360° consistency cannot be assessed.

4. Lack of Animation Capability

Competing methods like LHM support animation of reconstructed avatars, while IPRM is limited to static reconstructions, restricting its applicability.

**Questions:**

Please see Weaknesses.

---

### Official Review · Reviewer_SZmE · 2025-10-31

**Soundness:** 3
**Presentation:** 3
**Contribution:** 2
**Rating:** 4
**Confidence:** 4

**Summary:**

Reconstructing 3D digital human from single-view images is a hot topic. The common way of existing works directly uses 2D features for 3D reasoning. This work argues that this will cause challenges to preserve 3D identity. Thus, a novel method is presented where it first project 2D features to a SMPL-guided 3D space and construt sparse voxel representation and then a 3D reasoning module is designed to propagate features from visible to invisible. Experiments verified that the proposed method outperforms existing methods.

**Strengths:**

- the motivation is very clear and the proposed design is also reasonable.
- the visual results of the proposed method, as shown in Fig 4,  are obviously better than others especially for the identity.

**Weaknesses:**

Although the method seems reasonable to me, I have several concerns on the results:
- Among all examples in Fig 4, 8,9,10,  many of the input images look like rendered from 3D assets. So, why not just use in-the-wild images? This makes me doubt about the generalization ability of the proposed model.
- For some examples, like the middle one of fig 4, although IPRM produces better face, some details are missed. For example, the ropes of the hat are missed while both LHM and PShuman can produce those details. What are possible reasons?
- For some examples, such as the second one in Fig 8, the color also changes by IPRM (as seen the region of upper body). What are reasons?

It seems from the results that only face region produced by the proposed method shows obvious better quality. I am curious that if putting more atttention on the face region during the training of previous methods will also work （for example, adding an extra loss functions on face part）.

**Questions:**

No.

---

### Official Review · Reviewer_1HPG · 2025-11-01

**Soundness:** 3
**Presentation:** 2
**Contribution:** 1
**Rating:** 4
**Confidence:** 5

**Summary:**

The paper presents IPRM, a feed-forward framework that reconstructs photorealistic 3D clothed humans from a single image in ~0.6 seconds while preserving identity consistency. IPRM performs 3D token reasoning directly in SMPL-based sparse voxel space: Projects 2D identity features into 3D voxel space. Classifies voxels as visible (identity tokens) or invisible (to be reasoned). Uses visible tokens to infer invisible regions while explicitly preserving identity, which seems reasonable. The good peformance is shown in the evaluation.

**Strengths:**

1.The identity-aware 3D reasoning module(although I don't think it is reasoning) with visibility mask-based self-attention explicitly preserves visible identity tokens.

2.The 3D ID Adapter provides token-level guidance to prevent identity drift during decoding.

3.The paper includes extensive quantitative and qualitative comparisons on multiple datasets (THuman2.1, Synthetic Data, CustomHuman) with both 3DGS and mesh reconstruction.

**Weaknesses:**

1."REASONING" is overclaimed. I don't see clearly the reasoning part. While the overall framework is reasonable, individual components (sparse voxels, cross-attention for conditioning, SMPL priors) are adaptations of existing techniques. The main contribution is the integration rather than fundamentally new methods.

2.The method heavily relies on accurate SMPL estimation from the input image. The paper doesn't thoroughly analyze failure cases when SMPL estimation is poor or discuss robustness to SMPL errors.

3.The authors acknowledge that the sparse voxel representation limits fine detail reconstruction. This is a significant limitation for applications requiring high-fidelity details (e.g., facial wrinkles, clothing textures).

4.Most quantitative evaluations are on controlled datasets with ground truth. More extensive evaluation on truly in-the-wild images would strengthen the claims.

**Questions:**

Generalization: How well does IPRM generalize to: Extreme poses not well-represented in SMPL?
Very loose clothing that significantly deviates from body shape?
Occluded body parts?

Computational Breakdown: Can you provide a breakdown of inference time across different components (voxel projection, reasoning module, decoder)?

Qualitative Failure Analysis: Can you show and discuss failure cases to better understand the method's limitations?

---

### Note · Authors · 2025-11-14

**Comment:**

We would like to express our sincere gratitude to the reviewers for their thoughtful and constructive feedback. We genuinely appreciate the effort invested in evaluating our work and had hoped to further interact with the reviewers to improve the manuscript based on their insights. However, due to certain constraints, we regret to withdraw the submission at this time. We will carefully incorporate the reviewers‘ comments to strengthen the work moving forward.

**Withdrawal Confirmation:**

I have read and agree with the venue's withdrawal policy on behalf of myself and my co-authors.